# Anti-B-Cell-Activating Factor (BAFF) Therapy: A Novel Addition to Autoimmune Disease Management and Potential for Immunomodulatory Therapy in Warm Autoimmune Hemolytic Anemia

**DOI:** 10.3390/biomedicines12071597

**Published:** 2024-07-18

**Authors:** Mahija Cheekati, Irina Murakhovskaya

**Affiliations:** 1Morristown Medical Center, Morristown, NJ 07960, USA; mahija.cheekati@atlantichealth.org; 2Montefiore Medical Center, Albert Einstein College of Medicine, Bronx, NY 10461, USA

**Keywords:** autoimmune hemolytic anemia, B-cell-activating factor (BAFF), immunomodulation, belimumab, povetacicept, atacicept, ianalumab, tabalumab, blisibimod, hematology

## Abstract

Although rituximab is not specifically approved for the treatment of warm autoimmune hemolytic anemia (WAIHA), the First International Consensus Group recommends considering its use as part of the initial therapy for patients with severe disease and as a second-line therapy for primary WAIHA. Some patients do not respond to rituximab, and relapses are common. These relapses are associated with elevated B-cell-activating factor (BAFF) levels and the presence of quiescent long-lived plasma cells (LLPCs) in the spleen. A new group of immunomodulatory drugs, B-cell-activating factor inhibitors (BAFF-i), demonstrated efficacy in multiple autoimmune diseases and have the potential to improve WAIHA treatment outcomes by targeting B-cells and LLPCs. This article reviews the role of BAFF in autoimmune disorders and the currently available literature on the use of BAFF-directed therapies in various immunologic disorders, including WAIHA. Collectively, the clinical data thus far shows robust potential for targeting BAFF in WAIHA therapy.

## 1. Introduction

Autoimmune hemolytic anemia (AIHA) is a disease that results in the immune-mediated destruction of red blood cells (RBC) through autoantibodies or complement [1]. It is relatively rare, with an incidence of 1–3 per 10^5^/year in adults and a prevalence of 17:100,000 [2]. AIHA is heterogeneous in presentation and categorized as warm (WAIHA), cold (cold agglutinin disease (CAD)), cold agglutinin syndrome, and paroxysmal cold hemoglobinuria, or mixed. While 50% of WAIHA tends to be primary (idiopathic), the majority of cold antibody AIHAs are secondary to underlying autoimmune disorders, lymphoproliferative disorders, organ transplantation, infections, or solid tumors [3]. The mainstay of WAIHA treatment is limited to steroids as a first-line therapy followed by rituximab, splenectomy for refractory cases, and other immunosuppressive agents or monoclonal antibodies [1]. However, relapses and cases refractory to these therapies are common in AIHA. Less than 40% of patients remain in long-term remission, 50–60% remain corticosteroid-dependent, and >50% require second-line therapy [4,5]. This prompts a need for a better understanding of the mechanisms of refractory AIHA and investigation into alternative treatment options. In several autoimmune diseases, BAFF inhibition was associated with immunomodulation and improved disease control. A number of B-cell-activating factor inhibitors (BAFF-i) are under investigation, and one has been approved for use. In this review, we discuss the preclinical and clinical studies on BAFF-i, including the ongoing clinical trials for WAIHA, and review the existing literature for their use in other autoimmune disorders (Figure 1).

## 2. Current Warm Autoimmune Hemolytic Anemia Treatment

For the past few decades, the mainstay of WAIHA treatment has consisted of steroids, IVIG, rituximab, splenectomy, and immunosuppressants. Steroids are the first line of therapy, and while most patients respond in 2–3 weeks, prolonged taper is necessary due to the increased risk of relapse with rapid taper [6]. Rituximab is recommended as second-line therapy in steroid-refractory cases, as first-line therapy in severe cases, or as a corticosteroid-sparing strategy [7]. Rituximab is an anti-CD20 monoclonal antibody that non-specifically depletes B-cells, leading to generalized immunosuppressive effects. In the past decade, rituximab has become the second-line therapy of choice over immunosuppressants or splenectomy due to its relatively favorable safety profile and an 80% overall response rate in WAIHA with a 60% sustained response at 3 years [8,9,10]. However, toxicities are not uncommon, with an increased risk of infections, infusion reactions, and impaired vaccine responses [10,11,12]. In patients who do not respond to rituximab or who relapse after treatment, non-specific immunosuppressants such as azathioprine, cyclosporine, cyclophosphamide, mycophenolate, and danazol or splenectomy are used, as summarized in Table 1 [13]. However, these options have a long time to response, less favorable side effect profile, and are associated with an increased infection risk due to immunosuppression and various organ toxicities [13]. In addition, the data are limited, and there are no prospective studies evaluating the efficacy and safety of these agents. While splenectomy is effective in most patients, the drawback is the chronically increased infection risk, reserving this option for third-line therapy [8]. These current lines of therapies have a high risk of relapse/refractory disease due to various mechanisms of B-cell escape at different stages of development or autoreactive plasma cell formation, leading to the reconstitution of pathogenic antibody-producing cells post-treatment. A number of therapies are currently being investigated in refractory cases, including plasma cell-directed therapies, splenic tyrosine kinase (Syk) inhibitors, neonatal Fc receptor (FcRn) antibodies, and Bruton tyrosine kinase (BTK) inhibitors [14]. BAFF inhibitors are a novel class of agents that address potential mechanisms of B-cell escape and are under investigation in WAIHA for their potential to reduce the risk of relapse.

## 3. B-Cell-Activating Factor

B-cell-activating factor (BAFF), also known as B Lymphocyte Stimulator (BLyS) and APRIL (a proliferation-inducing ligand), are cytokines in the tumor necrosis factor (TNF) ligand family that are widely expressed by several cell types, including hematopoietic and stromal cells. BAFF and APRIL interact with the TNF receptor family B-cell maturation antigen (BCMA and transmembrane activator and CAML interactor (TACI). In addition, BAFF interacts with BAFF-R, a receptor specific for BAFF. In humans, BAFF-R is expressed at the late transitional (T1) B-cell stage in all mature B-cells and memory B-cells and is absent in plasma cells. TACI is expressed on B-cells after the T2 stage and on plasmablasts and plasma cells, while BCMA is exclusively upregulated on plasmablasts and plasma cells. BAFF is essential for B-cell development, maturation, and survival [15,16]. BAFF-deficient mice have unimpaired B-cell production in the bone marrow and naive T1 transitional B-cells in the spleen but impaired subsequent stages of B-cell differentiation, including T2, mature, and marginal zone B-cells. BAFF-R-deficient mice have B-cell lymphopenia and weak high-affinity antibody responses. In humans, BAFF-R deficiency was seen in association with adult-onset immune deficiency, B-cell arrest at the transitional B-cell stage, decreased numbers of mature B-cells, including follicular, marginal zone, and memory B-cell lymphopenia, decreased serum IgG and IgM, and normal IgA serum levels and impaired T-cell-independent vaccine immune responses [17]. 

BAFF excess in mice is associated with systemic disease that closely resembles human SLE and Sjögren’s syndrome, characterized by lymphoproliferation, excessive autoantibody production, increased numbers of peripheral B-cells, and hypergammaglobulinemia [18]. In autoimmune disorders, increased circulating BAFF rescues mature self-reactive B-cells from deletion, promoting the survival of autoreactive B-cells [19]. This leads to the expansion of autoreactive B-cells that are identified in many autoimmune disorders, including Sjogren’s syndrome, systemic lupus erythematosus (SLE), and rheumatoid arthritis (RA), making BAFF an attractive target for immunomodulatory disease therapy [20,21]. BAFF levels are also shown to be increased in WAIHA in positive correlation with disease activity and decreased with glucocorticoid therapy, suggesting BAFF’s pathologic role in WAIHA [6]. Furthermore, elevated BAFF levels and autoreactive plasma cells were observed in the spleens of WAIHA patients who underwent splenectomy due to rituximab treatment failure, implicating BAFF as a promising target in refractory WAIHA treatment [22].

## 4. Advantages of BAFF Inhibition for Relapsed/Refractory WAIHA Therapy

Rituximab is recommended in steroid-relapsed or refractory WAIHA, but relapses are seen in 30% of patients, and a subset of these patients may subsequently become refractory to rituximab [7]. Rituximab is a murine monoclonal antibody that binds CD-20 on B-cells, leading to antigen-dependent cell-mediated cytotoxicity (ADCC), complement-mediated cell lysis, and induction of apoptosis, resulting in non-specific B-cell depletion [13]. CD20 expression is limited to B-cells, from the pre-B stage to memory cells, while plasma cells (PCs) do not exhibit this marker, resulting in a population of PCs unaffected by rituximab. Incomplete B-cell depletion has been associated with a nonresponse to rituximab. Further, the rate of B-cell reconstitution post-rituximab is an indicator of the risk of relapse [23,24].

Analysis of splenectomy specimens of WAIHA patients who did not respond to rituximab at the time of B-cell depletion revealed a unique population of mutant, quiescent splenic B-cells, characterized by the downregulation of B-cell-specific factors and the expression of pro-survival genes. These characteristics facilitated escape from rituximab depletion. Once rituximab clears, autoreactive B-cells reactivate, giving rise to PCs and stimulating the generation of naive B-cells in germinal centers for reconstitution, leading to relapse [25]. WAIHA patients who were resistant to rituximab were also found to have long-lived plasma cells (LLPCs) in the spleen that produce antibodies facilitating disease relapse. The spleen cell cultures of these patients were found to have elevated levels of BAFF, prompting the conclusion that non-specific B-cell depletion by rituximab may promote the maturation and survival of autoreactive LLPCs in the spleen [22]. In mice, autoreactive B-cells seem to rely more on BAFF for their survival compared to non-autoreactive B-cells [19]. BAFF inhibition can preferentially target autoreactive B-cells with a lower risk of B-cell lymphodepletion.

A follow-up study combining anti-CD20 and anti-BAFF therapies showed a reduction in LLPCs, suggesting BAFF inhibitors can be a component of effective therapy to overcome rituximab resistance in WAIHA by acting on LLPCs that are left unaddressed by rituximab [26]. The serum of patients with AIHA and ITP have elevated levels of BAFF and APRIL compared to healthy individuals, and polymorphisms in BAFF are associated with ITP [27,28,29,30]. The BAFF inhibitor belimumab was found to be effective in combination with rituximab in ITP and as a monotherapy in SLE-associated ITP [31,32]. 

The pivotal role of B-cells in autoimmunity, combined with BAFF, established a mechanism of action involving the targeted depletion of B-cells, highlighting the need to investigate BAFF inhibition for treating autoimmune disorders. Given the significant role of B-cells in antibody production, BAFF is important in the pathogenesis of WAIHA. BAFF overexpression in WAIHA is associated with the breakdown of B-cell tolerance and increased autoantibody production. Importantly, BAFF is involved in B-cell repopulation following their depletion by rituximab in WAIHA, making it a target for preventing relapse through BAFF modulation.

Current BAFF-directed therapies that are undergoing clinical trials for use in autoimmune disease include belimumab, tabalumab (LY2127399), blisibimod (A-623), ianalumab (VAY736), atacicept, povetacicept (ALPN-303), rozibafusp alfa (AMG-570), and BAFF CAR-T. Table 2 lists the most notable trials for anti-BAFF agents in autoimmune disorders.

## 5. Belimumab

Belimumab is a recombinant fully humanized monoclonal IgG1 lambda antibody that binds soluble BAFF, preventing BAFF-R binding on B-cells, causing subsequent apoptosis and depletion of circulating B-cells [33]. Administration of belimumab antagonized BAFF-mediated increases in splenic B-cell numbers and IgA titers in murine models and was associated with B-cell depletion in the spleens and mesenteric lymph nodes in cynomolgus monkeys [33].

Belimumab is the first and only BAFF-i that is FDA-approved for use in SLE. It can be administered intravenously or as a weekly subcutaneous injection. BLISS-52 and BLISS-76 are phase III randomized controlled trials (RCTs), showing significant efficacy and safety of belimumab in SLE patients along with reduced rates of relapse, prednisone doses, improved serologic outcomes, and prolonged time until first flare, which were all maintained long-term in follow-up studies [34,35,36]. Studies of lupus nephritis demonstrate improved renal outcomes [37]. Multiple pooled data analyses of these trials support these findings, reflecting BAFF-dependent survival of B-cells in contrast to plasma cells [36,38]. Belimumab demonstrated a consistent safety profile in multiple phase II and III trials with a low incidence of adverse events similar to the placebo [39]. In fact, the rates of adverse events and infections over a 7-year period, covering approximately 1745 patient years, have either remained stable or decreased over time [40]. The most common side effects are infusion-related gastrointestinal reactions, with the rate of neutropenia being low (4%). No specific premedications or antimicrobial prophylaxis is recommended.

There are many active or recruiting trials assessing the efficacy of belimumab in other autoimmune diseases, including systemic sclerosis, RA, vasculitis, chronic lymphocytic leukemia (CLL), graft versus host disease, allogeneic transplant immunosuppression, neuromyelitis optica spectrum disorders, Sjogren’s syndrome, antiphospholipid syndrome, membranous nephropathy, myositis, and myasthenia gravis.

Belimumab has shown promising efficacy in the treatment of immune thrombocytopenia in the setting of systemic lupus erythematosus [31]. A single-center RITUX-PLUS trial evaluating belimumab with rituximab in 15 patients with persistent or chronic ITP demonstrated significant efficacy with an 80% overall response rate and 67% complete response at 52 weeks [32]. This suggests the role of BAFF inhibition in preventing the emergence of pathogenic LLPCs and a follow-up randomized placebo-controlled phase III trial is in the process of recruiting patients. Although belimumab had a minor effect on hematologic manifestations of SLE in a combined analysis of the two phase III trials [41], these findings in ITP patients are promising indicators for extending the investigation of belimumab to WAIHA.

## 6. Tabalumab

Tabalumab is a human IgG4 monoclonal antibody that binds both soluble and membrane BAFF, which can be administered subcutaneously [42]. In three preclinical immunotoxicology studies in cynomolgus monkeys, administration of tabalumab was associated with a sustained decrease in circulating B-cell populations but no change in follicular histology, with a reversible non-adverse microscopic decrease in the size of germinal centers in the spleen and lymph nodes; T-cell-dependent humoral immunity was preserved [43]. Studies in rabbits and cynomolgus monkeys also demonstrated no adverse reproductive or developmental effects [44,45].

Clinical trials for tabalumab, predominantly in RA and SLE, show mixed results. Randomized placebo-controlled trials in SLE patients, ILLUMINATE-1 and ILLUMINATE-2, both demonstrate biological activity, changes in anti-dsDNA, complement, B-cells, and immunoglobulins consistent with BAFF pathway inhibition. However, there were discrepancies in clinical outcomes, with the preliminary findings suggesting a lack of efficacy, leading to the termination of further studies in SLE [46,47].

In RA, four phase II trials evaluating the dose range of tabalumab consistently demonstrated efficacy and a good safety profile [48,49,50]. However, two phase III randomized placebo-controlled trials of subcutaneous tabalumab in RA patients failed to demonstrate an effect on disease activity despite significant decreases in CD3-CD20 B-cells and serum immunoglobulins [51,52].

Tabalumab was also evaluated in combination with bortezomib in relapsed multiple myeloma. While initial phase I trials suggested efficacy, a phase II trial showed no significant improvement in progression-free survival [53,54,55]. No active clinical trials with tabalumab are ongoing due to the lack of efficacy in most completed trials, while some show potential for use in RA only, indicating that this BAFF-i may not be generalizable to other autoimmune conditions.

## 7. Blisibimod

Blisibimod is a peptibody fusion between the Fc portion of IgG and high-affinity soluble and membrane-bound BAFF-binding peptides. However, the peptibody binding domain is synthetic, increasing the risk of the development of a neutralizing antibody that can reduce the effectiveness of blisibimod [56]. It was studied in phase II/III trials for SLE, IgA nephropathy, ITP, and vasculitis. Early studies in murine models for lupus and arthritis showed effective BAFF binding and inhibition with a subsequent reduction in B-cell numbers and disease manifestations in mice [57]. Phase I and II (PEARL-SC) trials of blisibimod (single dose and multiple doses) in SLE patients demonstrate a favorable safety profile [58,59]. A phase III trial of blisibimod in patients with SLE with high disease activity did not meet the primary endpoint of reduction in disease activity. Blisibimod administration was associated with successful steroid taper, reduction in proteinuria, and biomarker responses [60]. Trials for blisibimod in IgA nephropathy, ITP, and vasculitis were withdrawn due to lack of efficacy, and no active trials are currently being pursued.

## 8. Ianalumab

Ianalumab (VAY736) is a humanized afucosylated IgG1 kappa monoclonal antibody engineered to competitively inhibit the human BAFF-R, preventing the binding of BAFF and interrupting BAFF-R-mediated signaling in B-cells. Additionally, ianalumab enhances ADCC [61]. In preclinical mouse models of CLL, ianalumab demonstrated efficacy and improved survival when used in combination with ibrutinib, a BTK inhibitor [62,63]. Single and multicenter phase II trials in Sjogren’s syndrome found significant, sustained B-cell depletion and a well-tolerated dose-dependent decrease in disease activity [61,64].

Active trials are ongoing in SLE, Sjogren’s syndrome, RA, autoimmune hepatitis, and non-Hodgkin’s lymphomas [65]. There are currently three trials investigating the efficacy of ianalumab in ITP, including a phase II trial in patients previously treated with a corticosteroid and a thrombopoietin receptor agonist [66]. VAYHIT1 and VAYHIT2 are phase III multicenter RCTs in ITP as first-line therapy with corticosteroids and as second-line therapy with eltrombopag, respectively [67,68]. The efficacy of ianalumab in WAIHA patients who failed at least one prior therapy is being evaluated in a phase III RCT VAYHIA [69].

## 9. Atacicept

Atacicept is a TACI-R-IgG1-Fc chimeric fusion protein that has the capability of neutralizing both BAFF and APRIL. This mechanism of action uniquely allows it to deplete plasma cells in addition to B-cells. Atacicept was studied in SLE, RA, multiple sclerosis, IgA nephropathy, and optic neuritis. While initial studies in murine lupus models showed a positive disease response, studies in patients with SLE were terminated due to an increased rate of infections associated with a rapid decline in IgG levels [70]. Trials in multiple sclerosis and optic neuritis were also terminated due to worsened disease activity [71,72].

Trials in RA as a second-line therapy alone or in combination with rituximab (AUGUST-III) demonstrated a decrease in immunoglobulin and rheumatoid-factor levels but failed to meet the primary clinical efficacy endpoint [73,74,75]. Atacicept demonstrated a significant reduction in proteinuria in patients with IgA nephropathy and is currently being evaluated in a phase III trial in this disease [76,77].

## 10. Povetacicept

BAFF and its related cytokine, a proliferation-inducing ligand (APRIL), can act by binding transmembrane activator and calcium-modulating cyclophilin ligand interactor (TACI) in addition to BAFF-R and BCMA, their respective receptors. Povetacicept (TACI vD-Fc; ALPN-303) is an Fc fusion protein of a TACI domain that is engineered to more potently inhibit both BAFF and APRIL (variant TNFR domain, vTD) compared to wild-type (WT) TACI-Fc molecules in preclinical studies [78]. In vitro human lymphocyte assays, mouse disease models for arthritis and SLE, and cynomolgus monkey models demonstrated more potent BAFF/APRIL co-antagonism, lymphodepletion (including plasma cells and follicular T-helper cells), suppression of serum immunoglobulins (IgA, IgM, and IgG), decreased titers of antigen-specific antibodies, and reduced lupus disease activity compared to the WT TACI-Fc molecules [79]. Mouse models of AIHA demonstrated a reduction in antibody-secreting cells and anti-RBC autoantibody synthesis as well as reduced lymphocyte subsets, including T follicular helper (Tfh) cells, total B-cells, germinal center (GC) B-cells, and plasma cells in spleens of povetacicept-treated mice; this was coupled with an increased hematocrit and limited serum LDH elevation [78,80]. These results are promising given povetacicept’s unique mechanism of action of inhibiting both BAFF and APRIL, resulting in a downregulatory effect on multiple stages of developing B-cells from immature to plasma cells. This contrasts pure BAFF inhibition, which does not extend to plasma cells since they are preferentially controlled by the effects of APRIL binding TACI or BCMA.

Povetacicept pharmacokinetics and pharmacodynamics in healthy volunteers, studied in the RUBY-1 trial, demonstrated a dose-dependent decrease in circulating autoantibody and B-cell levels according to the initial reports by Alpine pharmaceuticals, though the full results have yet to be published [78]. Two other open-label trials are currently recruiting patients: one evaluating povetacicept in vasculitis, lupus nephritis, and autoimmune nephropathy (RUBY-3), and the other evaluating povetacicept in autoimmune cytopenias, including AIHA, ITP, and CAD (RUBY-4). Preliminary reports of the RUBY-4 study demonstrate a reduction in B-cells and immunoglobulin levels after the initial doses [81].

## 11. Rozibafusp Alfa

Rozibafusp alfa (AMG-570) is a first-in-class bispecific antibody-peptide conjugate that inhibits BAFF and inducible costimulator ligand (ICOSL). The ICOS/ICOSL costimulatory pathway is important for T-cell activation, migration, differentiation, and antibody production, mediated by interactions between follicular helper T-cells and B-cells. This unique mechanism of action blocks both B-cell maturation and T-cell activation, as well as aberrant B- and T-cell interactions, preventing downstream autoimmune effects, particularly in diseases such as SLE and RA. Rozibafusp alfa’s dual inhibitory effect was evaluated initially in mouse models for SLE and RA, cynomolgus monkeys, and human B-cell and T-cell assays. In the murine SLE and RA models, bispecific inhibition of both BAFF and ICOSL proved to be more efficacious than either target alone. Rozibafusp alfa binding to BAFF and ICOSL in cynomolgus monkey and human cell assays was also successful and was associated with B-cell depletion [82]. Single subcutaneous escalating doses in healthy volunteers were well-tolerated; respiratory tract infections were the most common adverse events. An overall reduction in naive B-cells and an increase in memory B-cells were observed with no effect on serum IgG or IgM [83]. A Phase Ib dose ascending study in patients with active RA demonstrated a reduction in naive B-cells, an increase in memory B-cells, and numeric improvement in disease activity in the higher-dose cohorts compared with the placebo. Treatment-associated adverse effects were seen in 96% of patients treated with AMG-570 and 88% of patients on the placebo and were mostly mild. Antidrug antibodies developed in 20% of patients with no apparent impact on safety [84].

## 12. BAFF and BAFF-R Chimeric Antigen Receptor T-Cell (CAR-T) Therapy

BAFF CAR-T therapy is unique from the other CAR-T targets in that inhibiting BAFF affects all three receptors that it binds downstream (BAFF-R, TACI, BCMA), covering potential B-cell escape pathways. Simultaneously, BAFF-R CAR-T is more selective for stages of B-cell development rather than generalized B-cell targets, such as CD19, which all B-cells express. Preclinical studies of BAFF CAR-T cells in cell assays and animal models of B-cell malignancies (mantle cell lymphoma, multiple myeloma, acute lymphoblastic leukemia) demonstrate robust cytotoxicity [85]. Another BAFF CAR-T construct demonstrated preclinical activity in non-Hodgkin’s lymphoma and acute lymphocytic leukemia cell lines, as well as tumor tissue from patients with CLL. It also had activity against CD19-deficient tumors [86]. A number of trials are currently investigating the role of BAFF-R CAR-T therapy in B-cell malignancies. An early phase I trial is currently evaluating the efficacy of CD19-BAFF CAR-T therapy in autoimmune diseases [87].

## 13. Conclusions

There is currently no approved therapy for WAIHA. While current first- and second-line therapies, steroids, and rituximab, respectively, have good efficacy, side effects are considerable, and many patients face relapsed disease. Third-line therapies place patients at risk of immunosuppression-associated consequences, and most lack prospective trials for their effectiveness. Multiple novel targeted therapies address various underlying immunological mechanisms of WAIHA. Relapse is attributed to mechanisms of autoreactive B-cell escape and LLPCs during treatment, followed by a reconstitution of the B-cell population. Due to this, BAFF inhibitors, particularly those that also inhibit APRIL, present a mechanism that targets all stages of B-cells. Thus far, BAFF inhibition has been approved for use in SLE and shows strong, positive evidence in clinical trials for other autoimmune diseases. Furthermore, some of the agents have been studied for use in conjunction with current therapies for diseases and demonstrated an improved treatment efficacy, and in some cases, even synergistically. Thus, anti-BAFF therapy holds promise for WAIHA as we await the final results of ongoing trials.

## Figures and Tables

**Figure 1 biomedicines-12-01597-f001:**
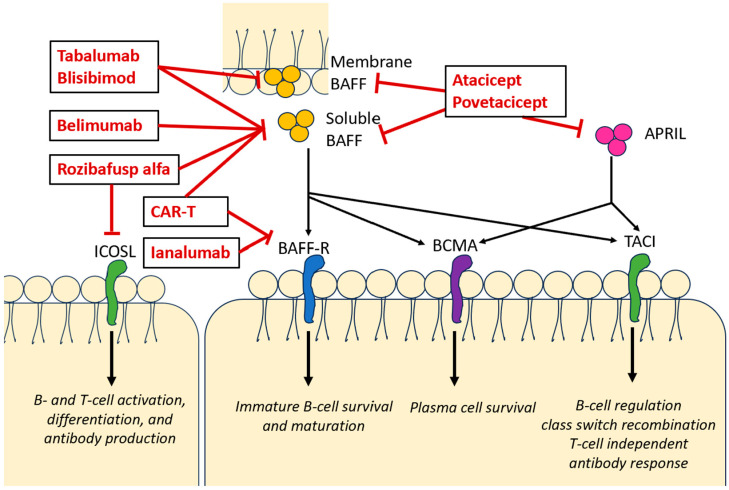
Anti-BAFF agents’ mechanisms of action.

**Table 1 biomedicines-12-01597-t001:** Currently used WAIHA therapies.

Line	1st	2nd	3rd					
Name	Steroids [6] [+IVIG]	Rituximab [7]	Azathioprine [13]	Cyclosporine [13]	Cyclophos Phamide [13]	Myco Phenolate [13]	Danazol [13]	Splenectomy [8,13]
Response Rate	70–80%	80%	60%	60%	50–70%	25–100%	20–50%	80%
Time to Response	2–3 weeks	3–6 weeks	1–3 months	1–3 months	2–6 weeks	1–3 months	1–3 months	1–2 weeks
Toxicities	-Weight gain -Hypertension -Diabetes -Peptic ulcer -Adrenal insufficiency -Psychosis -Myopathy -Insomnia	-Infusion reaction -Hypogammaglobulinemia -Neutropenia -Infections -Impaired vaccine response	-Immuno suppression -Myelotoxicity -Hepatotoxicity	-Immunosuppression -Hypertension -Renal injury	-Myelosuppression -Infection -Urotoxicity -Teratogenic -Infertility	-Immuno suppression -GI symptoms (nausea, diarrhea)	-Androgenic effect -Hepatotoxicity	-Infection, 50% mortality with immunization -Immunosuppression -Thrombosis
Comments	Slow taper required to reduce risk of relapse.	For steroid- refractory cases. First line for severe cases or when steroids are contra indicated	Retrospective studies and case reports			For pediatrics	Retrospective studies and case reports	Surgical risk is higher in older patients. No long-term data in WAIHA are available

IVIG: intravenous immunoglobulin; WAIHA: warm autoimmune hemolytic anemia.

**Table 2 biomedicines-12-01597-t002:** Notable clinical trials.

Drug/Disease/Mechanism of Action	Study Type	Results	Trial
**Belimumab** **Human IgG 1 against BAFF** **Targets Soluble BAFF**			
SLE	Phase I	Well-tolerated and reduced peripheral B-cell levels in SLE patients.	NCT00657007
	Phase II	Belimumab, in addition to standard therapy, can be well-tolerated in long-term use.	NCT00583362
	Phase III	Discontinuation of belimumab after treatment did not result in rebound SLE flares.	NCT02119156
	Phase II	Intervention with rituximab and belimumab led to reduced anti-nuclear antibodies and neutrophil extracellular traps.	NCT02284984
	Phase III	BLISS-76. Belimumab, in addition to standard therapy, improved disease response rates and reduced flares.	NCT00410384
	Phase III	BLISS-52. Significant improvement in disease symptoms and reduced number of flares in the belimumab-treated group versus placebo.	NCT00424476
	Observational	Pooled analysis of BLISS-52 and BLISS-76, confirming validity of results.	NCT01914770, NCT01858792
	Phase III	In hypocomplementemic, anti-dsDNA-positive SLE patients, belimumab improves disease symptoms, reduces flares, and reduces corticosteroid use.	NCT01484496
	Phase III	BLISS-BELIEVE. Adding a single cycle of rituximab to belimumab did not improve disease control/remission.	NCT03312907
Rheumatoid Arthritis	Phase II	Belimumab reduced disease symptoms, particularly in patients who were rheumatoid-factor positive or anti-citrullinated-antibody positive.	NCT00071812
ITP	Phase II	In combination with rituximab in patients with persistent or chronic ITP, the complete response was 66.7%–the overall response was 80%. The rate of B-cell repopulation was similar in rituximab + belimumab and rituximab groups, but T follicular helper cells significantly decreased in the belimumab group.	NCT03154385
	Phase III	Recruiting.	NCT05338190
**Tabalumab** **Human IgG4 against BAFF** **Targets soluble + membrane BAFF**			
SLE	Phase III	Terminated due to lack of efficacy.	NCT02041091, NCT0148870
	Phase III	ILLUMINATE-1, ILLUMINATE-2. Improvement in anti-dsDNA antibodies, complement, total B-cells, and immunoglobulins. ILLUMINATE-1. No difference in disease symptom improvement in treatment versus placebo groups and further secondary endpoints were not met. ILLUMINATE-2. Some secondary endpoints were not met—a rare side effect of suicidal ideation.	NCT01196091 NCT01205438
Multiple Myeloma	Phase II	No improvement in progression-free survival with tabalumab versus placebo.	NCT01602224
	Phase I	Efficacy in combination with bortezomib in relapsed-refractory multiple myeloma, but had significant adverse effects.	NCT01556438
	Phase I	Dose escalation and identification study saw full target neutralization obtained at a dose of 100 mg.	NCT00689507
Rheumatoid Arthritis	Phase II	Total B-cell counts decreased by ~40% from BL in treatment groups with improvement in disease symptoms.	NCT00837811
	Phase II	Efficacy was not achieved at the end of the trial period.	NCT00689728
	Phase II	The most significant disease response rates were observed in the 120 mg tabalumab group. Tabalumab treated patients suffered higher rates of infections.	NCT00785928
	Phase II	Tabalumab significantly reduced RA disease symptoms and serum IgM. B-cells were not completely depleted.	NCT00308282
**Blisibimod** **Peptibody against BAFF** **Targets soluble + membrane BAFF**			
SLE	Phase III	The primary endpoint of improvement in disease symptoms was not met, but blisibimod was associated with successful steroid reduction, decreased proteinuria, and biomarker responses.	NCT01395745
	Phase II	Significant disease response rates were noted in the high-dose blisibimod group.	NCT01162681
	Phase Ia	Blisibimod treatment decreased naïve B-cells (24–76%) with no changes in T-cells or NK cells.	NCT02443506
ITP	Phase II/III	Withdrawn. No study results.	NCT01609452
**Ianalumab** **Human IgG1 against BAFF** **Targets BAFF Receptor**			
SLE	Phase II, III	Recruiting.	NCT03656562, NCT0613397, NCT05624749, NCT 05639114 NCT05126277
Sjogren’s Syndrome	Phase II	Decrease in disease activity in ianalumab groups, with the biggest improvement in the 300 mg group.	NCT02962895
	Phase II/III	Recruiting.	NCT05985915, NCT05124925, NCT05349214, NCT 05350072
ITP	Phase II/III	Recruiting.	NCT05885555, NCT05653349, NCT05653219
Rheumatoid Arthritis	Phase I	Active trial.	NCT03574545
WAIHA	Phase III	Recruiting.	NCT05648968
**Atacicept** **TACI-R-IgG1-Fc against BAFF + APRIL** **Targets soluble + membrane BAFF**			
SLE	Phase II	Significant disease response rates to atacicept treatment were seen in subgroups with high levels of disease activity or serologically active disease.	NCT01972568
	Phase II	Significant disease response rates to atacicept treatment were seen in subgroups with high levels of disease activity or serologically active disease.	NCT01972568
	Phase II/III	Differences in flare rate or time to first flare in high-dose atacicept showed a significant reduction, but no difference in low-dose atacicept compared to placebo. Both atacicept groups show reductions in total Ig levels and anti-dsDNA antibodies and increases in C3 and C4 levels.	NCT00624338
	Phase II/III	Terminated due to adverse effects, including severe proteinuria and unacceptably low serum IgG.	NCT00573157
Multiple Sclerosis	Phase II	Reduced expression of rheumatoid factor and Ig, but no improvement in symptoms, failing to meet primary endpoint.	NCT00430495
	Phase II	Terminated due to increased MS disease activity in atacicept arms compared to placebo.	NCT00642902, NCT00853762
Rheumatoid Arthritis	Phase II	Atacicept combined with rituximab did not show clinical benefit in RA. Peripheral B-cells remained too low to determine if atacicept delayed B-cell repopulation after rituximab depletion.	NCT00664521
	Phase II	Disease response did not differ in the atacicept group from the placebo group.	NCT00595413
**Povetacicept** **TACI vTD-Fc against BAFF + APRIL** **Targets soluble + membrane BAFF and APRIL**			
Healthy Volunteers	Phase I	Reduction in circulating immunoglobulins and antibody-secreting cells.	NCT05034484
Vasculitis Nephropathy Nephritis	Phase I, II	Recruiting.	NCT05732402
AIHA/ITP	Phase I, II	Recruiting.	NCT05757570
**Rozibafusp alfa** **Antibody-Peptide against BAFF + ICOSL** **Targets soluble + membrane BAFF**			
Healthy Volunteers	Phase I		NCT02618967
SLE	Phase II	Completed without results.	NCT04058028
RA	Phase I		NCT03156023
**CAR-T** **against BAFF, BAFF-R, and CD19**			
Hematologic Malignancies	Phase I	Recruiting.	NCT06191887
Autoimmune Diseases	Phase I	Recruiting.	NCT06279923

## Data Availability

No new data were created or analyzed in this study. Data sharing is not applicable to this article.

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
