# Peer review of "Anti-B-Cell-Activating Factor (BAFF) Therapy: A Novel Addition to Autoimmune Disease Management and Potential for Immunomodulatory Therapy in Warm Autoimmune Hemolytic Anemia"

_biomedicines, 2024, doi:10.3390/biomedicines12071597_

Round 1

Reviewer 1 Report

Comments and Suggestions for Authors

In this current review “Anti-B Cell Activating Factor (BAFF) Therapy: The Promising Future of Immunomodulatory Therapy in Warm Autoimmune Hemolytic Anemia”, authors review how the different therapies can be used for the treatment for warm autoimmune hemolytic anemia, the study is informative and can be more refined and useful if the following points can be included:

1)Though authors have reported several treatment regimens for AIHA but hasn’t concluded the study about the different aspects of the study.

2)Authors should mention the Limitations and drawbacks of the different therapies

3)The review should include the future prospects for the study that will focus the relevance of writing this review.

4) Is there any combinatorial treatment that is more effective than using these monotherapies.

5)Authors should also mention about the management of these BAFF agents used for the treatment.

Author Response

In this current review “Anti-B Cell Activating Factor (BAFF) Therapy: The Promising Future of Immunomodulatory Therapy in Warm Autoimmune Hemolytic Anemia”, authors review how the different therapies can be used for the treatment for warm autoimmune hemolytic anemia, the study is informative and can be more refined and useful if the following points can be included:

Response: The authors appreciate the reviewer’s positive response to our review and the opportunity to further refine the presentation of its key points.

1)Though authors have reported several treatment regimens for AIHA but hasn’t concluded the study about the different aspects of the study.

Response: We thank the referee  for the revision and  recommendation. We have added a table (Table 1,  (Section 2. Current Warm Autoimmune Hemolytic Anemia Treatment, page 3)) more clearly summarizing the current lines of wAIHA therapies.  In addition, we added a conclusion section (section 13) which summarizes the key aspects of the paper.

2)Authors should mention the Limitations and drawbacks of the different therapies

Response: Thank you for the recommendation. We added a new table 1 (Section 2. Current Warm Autoimmune Hemolytic Anemia Treatment, page 3) summarizing current lines of wAIHA therapies including two rows about the toxicities and other notable comments.

3)The review should include the future prospects for the study that will focus the relevance of writing this review.

Response: We would like to thank the referee for pointing this out. Under section 2, we have added a sentence (Line 68-71) noting the relevance of the treatments discussed in this review, before we delve into the details of its value in section 4.

4) Is there any combinatorial treatment that is more effective than using these monotherapies.

Response: Thank you for the recommendation. Certain agents studied as combination therapy are discussed in their respective sections. A Conclusion has now been added to highlight the potential for synergistic use of some of these agents with existing therapies.

5)Authors should also mention about the management of these BAFF agents used for the treatment.

Response: Thank you for the recommendation. Lines have been added for belimumab on route of administration (Line 182), side effects (Line 192-193), and prophylaxis (Line 194).

Reviewer 2 Report

Comments and Suggestions for Authors

Cheekati and Murakhovskaya provide in the current review a comprehensive overview about the potential of Anti-B cell Activating Factor (BAFF) as a target in warm autoimmune hemolytic anemia (WAIHA). The authors have nicely summarized the role of BAFF in immune responses and the different ways how researchers develop therapeutic strategies against this important receptor and soluble factor.

Biologicals such as monoclonal antibodies are important tools nowadays against various autoimmune diseases. Several clinical trials have been and are currently performed. Table 1 gives an interesting summary and overview about the situation in the field. 

Yet, there is obviously only 1 clinical trial (phase III) in the recruiting phase where BAFF is targeted by Ianalumab, a human IgG1 antibody from Novartis which is directed against BAFF receptor. So, it is difficult to understand, why the authors wrote a review about WAIHA and targeting of BAFF? The finding that soluble BAFF is increased in WAIHA is nice data but not sufficient to name BAFF therapy the “promising future in immunomodulatory therapy in WAIHA.

So, the title is too forceful and the paper should better focus on BAFF in various autoimmune diseases and not so much on WAIHA.

Author Response

Cheekati and Murakhovskaya provide in the current review a comprehensive overview about the potential of Anti-B cell Activating Factor (BAFF) as a target in warm autoimmune hemolytic anemia (WAIHA). The authors have nicely summarized the role of BAFF in immune responses and the different ways how researchers develop therapeutic strategies against this important receptor and soluble factor.

Biologicals such as monoclonal antibodies are important tools nowadays against various autoimmune diseases. Several clinical trials have been and are currently performed. Table 1 gives an interesting summary and overview about the situation in the field.

Response: We would like to thank the Referee for revising our manuscript and for the positive feedback. We further edited the Table (now table 2) for conciseness

Yet, there is obviously only 1 clinical trial (phase III) in the recruiting phase where BAFF is targeted by Ianalumab, a human IgG1 antibody from Novartis which is directed against BAFF receptor. So, it is difficult to understand, why the authors wrote a review about WAIHA and targeting of BAFF? The finding that soluble BAFF is increased in WAIHA is nice data but not sufficient to name BAFF therapy the “promising future in immunomodulatory therapy in WAIHA.

So, the title is too forceful and the paper should better focus on BAFF in various autoimmune diseases and not so much on WAIHA.

Response: Thank you for the feedback. We appreciate the opportunity to adjust the title to more accurately reflect the utilization of BAFF inhibitors in autoimmune disease with potential for future use in wAIHA. Role of BAFF inhibitors in WAIHA is being evaluated in two Phase 3 clinical trials: one with Ianalumab, a BAFF-R antibody, and another with Povetacicept, a BAFF/APRIL antibody.  It should be noted there is no therapy currently approved for WAIHA and the number of clinical trials is limited due to the rare nature of the disease.  We made some additional changes to the manuscript that we hope you find useful.

Round 2

Reviewer 2 Report

Comments and Suggestions for Authors

The revisions and changes made by the authors have clearly increased the quality of the paper and the now give additional information about targeting of BAFF in autoimmune diseases. The potential use of BAFF therapies is now presented not so forceful compared to the initial version. Thereby the paper is now suitable for publication in Biomedicines.